# Nutritional, Microbial, and Sensory Evaluation of Complementary Foods Made from Blends of Orange-Fleshed Sweet Potato and Edible Insects

**DOI:** 10.3390/foods9091225

**Published:** 2020-09-02

**Authors:** Isaac Agbemafle, Doris Hadzi, Francis K. Amagloh, Francis B. Zotor, Manju B. Reddy

**Affiliations:** 1Department of Food Science and Human Nutrition, Iowa State University, Ames, IA 50011, USA; 2School of Public Health, University of Health and Allied Sciences, Hohoe PMB 31, Ghana; hadzi.doris@yahoo.com (D.H.); fbzotor@uhas.edu.gh (F.B.Z.); 3Department of Food Science and Technology, University for Development Studies, Tamale TL 1350, Ghana; fkamagloh@uds.edu.gh

**Keywords:** acceptability, cricket, microbial, nutrients, palm weevil larvae, sensory, weanimix

## Abstract

Improved formulations of complementary foods (CFs) with animal-source foods (ASFs) is an important strategy to improve infant and young child feeding (IYCF). However, ASFs are expensive in many food-insecure settings where edible insects abound. CFs were developed from flours of orange-fleshed sweet potato (OFSP) and cricket (OFSCri) or palm weevil larvae (OFSPal) or soybean (OFSSoy) in the ratio 7:3. Nutritional and microbial quality of the novel CFs were determined and compared with Weanimix (recommended maize-peanut-soybean blend). Sensory evaluation of porridges was rated on a five-point hedonic scale among 170 Ghanaian mothers. OFSCri (20.33 ± 0.58 g/100 g) and Weanimix (16.08 ± 0.13 g/100 g) met the protein requirement of 15 g/100 g from CFs. Although Fe content was significantly higher for OFSCri (1.17 ± 0.03 mg/100 g), none of the CFs met the recommended levels for Fe. All the CFs were free from *Salmonella*, and aerobic plate count was significantly below permissible levels. All the CFs were ranked above the minimum threshold (hedonic scale = 3; neither like nor dislike) of likeness for the sensory attributes. Crickets and palm weevil larvae can be blended with OFSP and could be sustainable, culturally appropriate alternative ASFs for IYCF, but long-term studies are needed to evaluate their efficacy.

## 1. Introduction

Stunting (low height-for-age) and wasting (low weigh-for-height) are common consequences of protein deficiency, whiles iron, zinc and vitamin A are the major micronutrient deficiencies negatively influencing growth during the complementary feeding period [1]. Complementary feeding starts with the timely introduction of safe and nutritious foods when breastmilk is no longer sufficient by itself, and this period ranges from 6–24 months of age. Globally, 140 million children under five years are stunted, and wasting still impacts the lives of over 47 million children, especially during the complementary feeding period [2]. Estimates indicate that 273 million children under five years are anemic and 5.2 million preschool-age children are at risk of night blindness from vitamin A deficiency (VAD); these estimates are skewed towards children from Africa and Asia [3,4]. Stunting has been associated with delayed motor development, poor cognitive and school performance, impaired social productivity and reduced income and livelihoods in adult life [1,5]. Anemia also has negative effects on physical performance and health status and thus increases healthcare costs [5]. Zinc deficiency is also associated with stunted growth, anemia, greater susceptibility to infection and other serious health consequences [1]. However, children from developing countries such as Africa widely consume plant-based complementary foods (CFs). These plant-based CFs are mainly from cereals (e.g., corn, millet, rice, or wheat porridge) and have low protein and micronutrient density, contributing to growth faltering and undernutrition in children.

To achieve protein and micronutrient density, complementary feeding guidelines recommend the addition of animal-source foods (ASFs) to plant-based CFs [6]. ASFs do not only increase nutrient density of CFs, but also improve bioavailability of micronutrients [7,8]. For example, ASFs not only provide bioavailable iron but also enhance the bioavailability of non-heme iron from other foods in the diet [7]. Unfortunately, the commonly recommended ASFs are expensive for low-income households, but using edible insects offer a promising alternative. Edible insects abound and have always been a part of the human diet, but are now gradually “rediscovered” for human consumption [9]. This is based on recent efforts since 2013 by the Food and Agriculture Organization (FAO) in promoting the eating of insects as a strategy to alleviate food and nutrition security by 2025 [9]. Edible insects are cheap, sustainable and locally available ASFs that can be used to improve the nutritional value of household foods in resource-poor communities [9]. Unlike edible insects, livestock production contributes to massive land and water degradation, leading to direct and indirect increases in greenhouse gas emissions [10]. Besides its lower environmental impact compared to ASFs from livestock, edible insects provide monounsaturated and polyunsaturated fatty acids, protein with comparable biological value to meat and fish, and even for some insects, significantly higher levels of iron than beef [11,12].

The World Health Organization (WHO) also recommends sustainably reducing childhood malnutrition by using available indigenous plants (e.g., Amaranth, cowpea, moringa, fonio, baobab, African eggplant, mushroom, etc.) and animal foods (e.g., small fish, eggs, rodents, etc.) to prepare CFs that are both hygienically and nutritionally adequate [6]. Besides WHO, the Codex Alimentarius commission that sets standards for various products have also specified recommended levels of nutrients for CFs [13]. Currently, orange-fleshed sweet potato (OFSP) is being promoted in sub-Saharan Africa as a food-based strategy to improve vitamin A content of CFs and to prevent VAD [14]. A blend of OFSP and soybean CFs, denoted OFSSoy, developed by other researchers do provide adequate protein and vitamin A, but not zinc and iron contents [14]. In addition, OFSSoy is not a better source of bioavailable iron compared with Weanimix [15]. We therefore hypothesized that adding edible insects to OFSP would improve its iron content and quality compared to Weanimix.

Weanimix is the mainstream recommended blend of maize, peanut, and soybean for infant and young child feeding (IYCF) in Ghana. In developing countries including Ghana, many households particularly in rural, peri-urban and urban-poor areas cannot afford ASFs to boost the nutrient density of complementary and household foods, but edible insects such as house cricket (*Acheta domesticus Linnaeus*) and palm weevil larvae (RFA, *Rhynchophorus phoenicis Fabricius*) have contributed to the diets of many adults in these deprived areas. For example, RFA (locally known as “*Gbamedo*” in the Ewe language in Ghana) is top among commonly consumed insects in southern Ghana [16] and is considered a delicacy [17]. Crickets (locally known as “*Ebore*” in the Ewe language in Ghana) are among the nine major insect species consumed in Ghana [16]. Based on our previous study in Ghana that reported willingness of caregivers to include edible insects into children’s food [17], CFs were developed using edible insects. The objectives of this study were to determine the nutritional value, microbial quality and mothers’ acceptability of CFs made from blends of OFSP and edible insects in comparison with Weanimix.

## 2. Materials and Methods

### 2.1. Food Samples and Chemicals

House cricket and RFA were respectively obtained from Aspire Food Groups Center in Austin, Texas, USA and Kumasi, Ghana and processed as described by Agbemafle et al. [18], with slight modifications. The processed insects obtained were steam cooked for 5 min and dried in a locally manufactured mechanical drier for 24–48 h. Unlike conventional cooking, steam cooking for 5 min was done to remove some of the fat from the insects, reduce its microbial load, and preserve its nutritional value, color, and flavor. Drying for 24–48 h was aimed at reducing the moisture content and microbial load. Subsequently, they were filtered (using a locally manufactured mesh of size 4.20 microns) to a flour of fine granularity and uniform consistency and stored in labeled Ziploc bags (Figure 1). OFSP storage roots were purchased from Kokubila Nasia Farms Ltd. (Tamale, Ghana). The roots were sorted, washed, peeled, diced, washed twice, steam cooked for 5 min and dried in a locally manufactured mechanical drier for 24–48 h. A hammer mill (Bravo Grinder, Washington, IA, USA) was used to grind the dried OFSP samples into flour. The flour was further sieved using a locally manufactured mesh (size 4.20 microns) to obtain a fine flour that was packaged in labeled Ziploc bags and stored at room temperature in a dry airy room.

The other ingredients (Table 1) were purchased from local markets in Accra, Ghana. These ingredients were chosen based on the nutritional requirements of infant and young children, the dietary habits of the population and the availability of these ingredients in Ghana. OFSP-cricket or -RFA blends were developed based on the current dietary reference value of protein, 11 g/day for infants 7–12 months [19]. The proportion of the different foods in each formulation was based on a previous study [14] and the intention to achieve 11 g/day protein [19]. The OFSP + cricket (OFSCri) and OFSP + RFA (OFSPal) were processed as shown in Figure 1. OFSP + soybean (OFSSoy) and Weanimix were processed as described by Amagloh and Coad [14]. All reagents used for laboratory analysis were analytical grade obtained from Sigma-Aldrich, St. Louis, MO, USA. Food production was carried out in the Food processing laboratory, Food Research Institute, Accra, Ghana.

### 2.2. Proximate Composition of the CFs

The methods described in the Official Methods of Analysis of the Association of Official Analytical Chemists (AOAC) International [20] were used to determine the amount of moisture in all the CFs (AOAC Method No. 925.10, slightly modified by drying the samples at 108 °C overnight for approximately 16 h). Crude protein (N × 6.25) was determined by the Kjeldahl method (AOAC Method No. 978.04). Crude fat was determined by exhaustively extracting a known weight of each CF in petroleum ether (boiling point, 40–60 °C) in a Soxhlet extractor (AOAC Method No. 930.09). Ash was determined by incineration (550 °C) of known weights of each CF in a muffle furnace (AOAC Method No. 930.05). Crude fiber was determined by the enzymatic-gravimetric method (AOAC Method No. 930.10). The carbohydrate content was determined by difference (i.e., by subtracting the sum of all the percentages of moisture, fat, crude protein, ash, and crude fiber from 100%). A bomb calorimetry (Parr 6200 calorimeter, Parr Instrument Company, Moline, IL, USA) method was used to measure the energy content of the CFs [20]. All the analysis was done on two independent replicates.

### 2.3. Mineral Analyses of the CFs

Mineral concentrations of the CFs were determined using standard procedures [20]. Briefly, the samples were ashed at 550 °C, then boiled with 10 mL of 20% hydrochloric acid in a beaker and filtered into a 100 mL standard flask and topped up to the mark with deionized water. Potassium (K) was determined from the resulting solution using Spectra AA 220FS flame emission photometer (Varian Co., Mulgrave, Australia) with an acetylene flame [20]. Calcium (Ca), iron (Fe), Magnesium (Mg) and Zinc (Zn) were also determined from the resulting solution using the atomic absorption spectrometry method [20]. A Perkin 400 atomic absorption spectrometer (Perkin Elmer Analyst 400, Waltham, MA, USA) with an air/acetylene flame and respective hollow-cathode lamps were used for absorbance measurements. Wavelengths, slits and lamp current used for the determination of the four elements were: 213.9 nm, 0.5 nm, 4.0 mA (Zn); 422.7 nm, 1.2 nm, 4.0 mA (Ca); 248.3 m, 0.2 nm, 6.0 mA (Fe) and 766.5 nm, 0.8 nm, 4.0 mA (Mg). The results for mineral contents were expressed as mg/100 g dry weight (DW).

### 2.4. CFs Contribution to Recommended Dietary Allowance (RDA) of Children

Percentage contribution to recommended dietary allowance (RDA) of children aged 6–12 months was expressed as a percent of the RDA.
(1)$$\%\mathrm{RDA}=\frac{X}{Y}\times 100$$
where *X* is the amount of nutrient analyzed and *Y* is the RDA for a given nutrient. 

### 2.5. Microbial Safety Testing of the CFs

In the microbial analyses, the total microbial count and testing for the presence of Enterobacteria, *Staphylococcus aureus*, *Salmonella*, *Shigella*, yeast and fungus were carried out. Specifically, each of the CFs was plated on one of the following culture media: MacConkey, cellulose with blood, Hektoen and Mannitol salt culture (OXOID Laboratories, Basingstoke Hampshire, England), and the media were incubated at 37 °C for 24 h. All recovered colonies in the media were suspended in the following agar: citrate of Simmons (for differentiation of Enterobacteria), Kliger and Mannitol to selectively and differentially test for the presence of Enterobacteria and *Staphylococcus aureus*. After incubation, we isolated fungus on Sabouraud cellulose agar. All microorganisms were identified based on colony morphology.

### 2.6. Maternal Acceptability of the CFs

#### 2.6.1. Study Design, Area and Recruitment

A cross-sectional design was used for the maternal sensory evaluation of the products to obtain first-hand information about mothers’ acceptance of the CFs for IYCF. The study was conducted among mothers attending child welfare clinics (CWCs) in Ho and Hohoe Municipalities in the Volta region, Ghana. A sample of 170 mothers were randomly selected from mothers presenting children for weighing at CWCs. On the CWC days, nurses in charge of the session, assisted by a trained research assistant, referred every mother aged 18–49 years who visited the center to the study enrolment team.

Referred mothers were given details of the study and were invited to participate. After consent, trained research assistants completed a screening questionnaire to determine if a potential mother had to be excluded because of illness or intolerance to milk or peanuts. Mothers who remained eligible after the screening were invited to participate in the sensory evaluation. The nurses were told to stop referring mothers after the target sample size for each CWC had been achieved. The study protocol was approved by the Institutional Review Board of the University of Health and Allied Sciences (UHAS–REC A.11 [5] 18–19) and Iowa State University (ISU IRB: 19–263–00) as well as the ethics review committee of the Ghana Health Service (GHS–ERC 013/07/19). Written permission to carry out the study was also obtained from the municipal/district health administration and from the CWCs.

#### 2.6.2. Maternal Sensory Evaluation

Maternal sensory evaluation involved hedonic ranking of the four CFs (one at a time). The CFs were prepared as porridges by the research team prior to the ranking. Each of the porridges was prepared by mixing 33 g of dry powder with 200 mL of water, then it was allowed to boil for about 5 min to a puree consistency (a consistency that does not readily fall off from a spoon). If necessary, additional water was added to obtain the puree consistency. The prepared porridges were cooled and then served to the mothers in 25 mL cups one at a time without weighing. Tasting of the porridges was carried out in temporal sensory booths at the CWCs under broad daylight (from 8 am to 12 pm). Prior to sensory evaluation, mothers were blinded to which porridge they were tasting, although they were informed all the CFs were made from OFSP and edible insects. Mothers were told that they were not expected to eat the whole amount provided. Between foods, the mothers were asked to rinse their mouth out with water. Mothers were told to rank the porridges for five attributes: appearance, aroma, taste, texture/mouth feel, after taste, and overall acceptability based on a five-point hedonic scale (“1”—dislike very much to “5”—like very much). Mothers were asked to record their responses on a paper-based questionnaire. These instructions were read directly to the mothers by trained research assistants after an informed consent was obtained from each of the 170 mothers who participated in the sensory evaluation. A score of 3 = neither liked nor disliked was considered the threshold for acceptance of the food [21]. All data collection forms for the sensory evaluation were checked by the supervisors in the field to allow immediate re-visits for gross errors or missing data.

### 2.7. Statistical Analysis

Nutrient composition of the CFs as well as the sensory attributes were presented as means with standard deviation. Differences in nutrient composition and the mean sensory attributes of the CFs were evaluated using one-way ANOVA followed by Tukey’s multiple comparison tests with significance set at *p* < 0.05. Microbial data were presented as counts. Data analysis were done using GraphPad Prism version 8 (La Jolla, CA, USA).

## 3. Results

### 3.1. Nutrient Composition of the OFSP-Based CFs and Weanimix

The nutrient composition (dry weight basis) for all the formulations are presented in Table 2. Except for OFSCri which met 95% of the specified energy (400 kcal) in the Codex standard, all the other CFs met this requirement (Table 2). OFSPal met 61% and OFSSoy met 80% of the protein specification of the Codex standard. OFSCri and Weanimix met the protein specification of the Codex standard for CFs for IYCF.

It was evident that there were significant differences between energy, carbohydrate, protein, and fat contents among the CFs. The energy content of the OFSP-based edible insect CFs were significantly lower compared to OFSSoy and Weanimix. Crude protein (20.33 g/100 g) was highest for OFSCri and significantly different from OFSSoy (11.95 g/100 g), Weanimix (16.08 g/100 g) and OFSPal (9.22 g/100 g, Table 2). The fat content was significantly lower for the OFSP-based edible insect CFs but was highest for OFSSoy (17.39 g/100 g, Table 2).

The ash content of OFSP-based CFs ranged from 6.36 to 6.78 g/100 g and this was significantly higher compared to Weanimix (1.91 g/100 g, Table 2). None of the CFs met the recommended levels of Fe and Zn for breastfeeding infants. OFSCri met 26% of the recommended Fe levels and <10% of the recommended Zn levels, while Weanimix met 17% and 12% of the recommended Fe and Zn levels. OFSSoy and OFSCri met the recommended Ca level from CFs for breastfeeding infants.

Table 3 shows the contribution of OFSP-based CFs and Weanimix to the RDA for energy, carbohydrate, protein, fat, Ca, Fe, and Zn for children aged 6–12 months. The CFs contributed the highest protein and lowest fat in reference to the RDA. The CFs contributed 83.8%–184.8% for protein but only 4.1%–58.0% of the RDA for fat for children in this age group (Table 3). Among all the CFs, the OFSP-based edible insect CFs contributed the highest to the RDA for Fe (10.7%), while Weanimix contributed the highest to the RDA for Zn (6.3%). The contribution of OFSCri to the RDA for Fe (10.6%) was about three times the contribution by OFSSoy (3.5%) and 1.5x the contribution by Weanimix (7.0%, Table 3). The contribution of OFSSoy to the RDA for Zn (1.7%) is about one-quarter the contribution to the RDA for Zn by OFSCri (4.3%) and OFSPal (4.7%).

### 3.2. Microbial Safety of the OFSP-Based CFs and Weanimix

The OFSP-based CFs and Weanimix were free from *Salmonella*, Enterobacteria, *Bacillus cereus*, yeast or fungus (Table 4). Mold was only present in OFSPal (30 cfu/g). Aerobic plate count (APC) was highest for OFSSoy (127 × 10^3^ cfu/g) and lowest for Weanimix (1 × 10^1^ cfu/g). APC for OFSCri and OFSPal were (38 × 10^1^ cfu/g) and (89 × 10^2^ cfu/g), respectively. Results of the microbial quality test of the CFs were within acceptable limits of the Ghana Food and Drugs Authority standards for dried products requiring heating to boiling before consumption (Table 4).

### 3.3. Sensory Attributes of the OFSP-Based CFs and Weanimix 

Table 5 presents results from the mean sensory attribute scores of porridges from OFSP-based CFs and Weanimix. Appearance was the most liked sensory attribute while taste was the least liked sensory attribute. Porridge from OFSSoy had the highest overall acceptability, but this was not significantly different from Weanimix. All the sensory attributes and overall acceptability were ranked lower for OFSP-based edible insect CFs porridge compared to OFSSoy and Weanimix (*p* < 0.05). Between the two OFSP-based edible insect CFs, OFSPal was ranked higher for all the sensory attributes (*p* = 0.001).

For all the CFs, mothers who liked a little or liked a lot were more than those who disliked a lot or disliked a little (Figure 2). In terms of overall acceptability, the proportion of mothers who liked a lot the porridge from OFSSoy (65.9%) or Weanimix (68.8%) was significantly higher than those who liked a lot the porridge from OFSCri (27.6%) or OFSPal (37.3%). It was observed that majority of the mothers liked the CFs very much in terms of the appearance, aroma, aftertaste, and texture (in decreasing order of sensory attribute likeness).

The analysis of variance showed that the overall acceptability of the CFs was found to be significantly dependent on the appearance (*p* < 0.05). Contrasting the appearance in terms of the color of the CFs is shown in Figure 3. The color of OFSPal was the brightest (bright yellow). Based on the appearance ranking by the mothers, it appears that the light-yellow color of OFSSoy and off-white color of Weanimix were most appealing to them.

## 4. Discussion

### 4.1. Orange-Fleshed Sweet Potato Based Complementary Foods (CFs) and Weanimix

Compared to Weanimix, OFSP-based CFs are nutritious (rich in vitamin A), naturally sweetened and safer for consumption [14]. More importantly, the addition of soybean to OFSP-based CFs guarantees the addition of energy, high-quality protein and micronutrients such as copper, folate, phosphorus and vitamin K [14]. Just like all plants, soybean is high in phytate that impairs iron and zinc bioavailability [8]. Most importantly, edible insects are ASFs that have little-to-no phytate levels and their addition to OFSP-based CFs provides energy, high-quality protein, and bioavailable iron and zinc among a host of other micronutrients [9,11,12]. Although OFSSoy and Weanimix were evaluated previously [14], the nutritional, microbial and sensory attributes of OFSCri and OFSPal for these novel CFs have not been studied.

### 4.2. Nutrient Composition

Processing dried CFs is important because the moisture content, which is an index of water activity and a measure of stability and susceptibility to microbial contamination, has a direct association with the shelf life of the CFs [13,14,21]. The moisture content of OFSP-based edible insect CFs as well as OFSSoy and Weanimix were below the moisture content (<5%) recommended by Codex standards for CFs and the critical moisture content (12%) for flours [13]. The low moisture content of these CFs can be attributed to the proper drying and handling during the processing of the foods. The CFs produced would be stable on the shelves for longer periods due to their low moisture content. On the other hand, a moisture content ranging from 2.4%–7.3% in different CFs produced from OFSP has previously been reported [14]. This implies that the OFSP-based CFs in this study would be more shelf-stable than those previously reported if they are packaged and stored under appropriate conditions [14,21]. Notably, the difference in processing methods could explain the variations in the moisture content. In this study, very low cost and a simple mechanical drying method was used whereas the previous study [14] used extrusion cooking, roller-drying and oven-toasting methods which are expensive and not common at the household level.

The significant differences between the energy content of OFSP-based CFs and Weanimix could be explained by the different ingredients in each CF. The higher energy content of OFSPal, OFSSoy and Weanimix compared to the Codex standard may be explained by the high fat content of these CFs. The fat contained in these CFs are mainly monounsaturated and polyunsaturated fatty acids which are good for brain development in infant and young children. The carbohydrate content of the OFSP-based edible insect CFs was also adequate to provide enough energy from carbohydrates to spare protein for growth. Based on the 1998 WHO estimated energy required from CFs [22], children could eat the following amounts of the OFSP-based porridges: 75 g/meal at 6–8 months, 120 g/meal at 9–11 months, and 200 g/meal at 12–23 months. These quantities compare well with average amounts usually consumed by children in the different age groups, but this may be influenced by the child’s appetite, the caregiver’s feeding behaviour and the characteristics (e.g., energy density and level of sweetness) of the porridges themselves [22].

The differences in the protein content of OFSCri and OFSPal compared to the protein value (15%) stipulated in the Codex standard for CFs [13] could be due to differences in protein content of cricket (20.1 g/100 g) and RFA (10.5 g/100 g) [12,24]. However, the protein content of OFSPal could be improved by using defatted RFA (dry basis) which is reported to have a protein content of 66.3% [25]. Although the protein content of OFSPal was below the codex standard, it could still make a good addition to the diet of children because its principal ingredient (RFA) is a good source of high quality and digestible protein. It also contains essential amino acids such as leucine and histidine [25] for enhanced growth in young children, making OFSPal suitable for IYCF as shown in an animal study in which RFA fed to weanling malnourished rats improved body composition and protein status [18]. Given its high protein content, about 30 g of OFSCri could be consumed to meet the 3–4.5 g of protein per daily ration recommended for infants 6–11 months [22]. This ration would need to be increased with age due to an increase in the body’s need for protein during growth in children.

The fat content of OFSCri was lowest because in crickets, the principal ingredient is reported to have low-fat content (5.1%) compared to RFA which contains 25.3% fat [12]. The low-fat content of OFSP-based edible insect CFs may be better for longer storage of OFSCri and OFSPal if properly packaged and stored in areas with low humidity and temperature [25]. The crude fiber content of OFSP-based CFs was significantly lower compared to Weanimix, a finding collaborating a previous study on sweet potato-soybean blend as an alternative CF to Weanimix for IYCF in Ghana [14]. The crude fiber content of the OFSP-based CFs was within the recommended Codex Standards (5%) for CFs [13], suggesting it’s suitable for complementary feeding. 

The high ash, specifically Fe and Zn contents, of the OFSP-based edible insect CFs compared to OFSSoy and Weanimix can be attributed to the fact that the edible insects are high in micronutrients [11,12]. For example, the Fe content of cricket and RFA is 12.1 mg/100 g and 2.58 mg/100 g, respectively [11,12]. Consuming 100 g of the OFSP-based edible insect CFs would not meet the RDA of 11 mg of non-heme Fe for infants 6–11 months old. As mentioned by other researchers [26], less dietary Fe is necessary from ASFs that have heme-associated Fe because ~30% is bioavailable as compared to ~10% bioavailability from non-heme Fe. Heme Fe in crickets and RFA could primarily be in the cytochromes, and its bioavailability is presumably similar to the heme Fe of myoglobin and hemoglobin [27]. Typically, Fe in insects are bound to ferritin and holoferritin in the ferrous state, which may have higher bioavailability [27]. It is therefore likely that the bioavailability of Fe in OFSCri and OFSPal CFs may be similar to other ASFs and superior to OFSSoy and Weanimix. Unlike OFSP flours which are low in phytate (0.19 g/100 g) [14], Weanimix was shown to have high phytate (0.48 g/100 g) and this would limit Fe and Zn bioavailability [8].

Although Zn is critical for cellular growth, none of the CFs met the recommended level of 4–5 mg of Zn in CFs for IYCF. Compared to a caterpillar cereal that provided 3.8 mg Zn daily [21], this OFSP-based edible insect CFs provided less because the proportion of OFSP flour to insect flour was 7:3 compared to 1:1 in the caterpillar cereal. Although we experimented with a 1:1 formulation, it was not used because it had the lowest score for all sensory attributes for a descriptive evaluation conducted among 30 untrained working mothers in the University of Health and Allied Sciences and University of Ghana (unpublished data). Thus, care must be taken to gradually introduce mothers and children to these OFSP-based insect CFs before deciding to increase the proportion of edible insects. In increasing the proportion of edible insects in these novel CFs, it is also important to consider its effect on the physico–chemical and functional properties of the CFs.

### 4.3. Microbial Safety

Generally, there are lots of concerns about the use of edible insects as food for humans because they can harbor *Salmonella* spp., *Campylobacter*, and *Staphylococcus* spp., fluke foodborne, waterborne pathogens and chemical hazards [9,12]. Our microbial quality test of the OFSP-based edible insect CFs showed that they are safe for consumption by human adults and children. This is in agreement with a previous study that showed that caterpillar cereal is safe for maternal and child consumption [21]. Comparing our microbial quality test to the Ghana Food and Drugs Authority standards, it meets safety standards for dried products requiring heating to boiling before consumption.

### 4.4. Sensory Evaluation

Sensory evaluation of a food product predicts its acceptability and the feasibility of introducing the food within a target population [28]. Maternal acceptability was crucial to provide an appropriate socio-cultural framework for the introduction of the OFSP-based edible insect CFs into infants’ diets. Although a previous study [28] on acceptability had focused on comparing new products to existing ones, other researchers have argued that it may be unreasonable [21]. In agreement with Paul et al. [28], the sensory evaluation in this study was designed to assess the acceptability of OFSSoy and its variations with Cricket or RFA in comparison with the mainstream CF, Weanimix. This is because OFSCri and OFSPal are novel CFs and, as such, requires comparison to existing CFs if they would eventually compete with these products on the Ghanaian market or be used in nutrition programs or for complementary feeding at the household level.

Overall acceptability was highest for porridges from OFSSoy and Weanimix, probably due to the familiarity of taste, aroma, and appearance with these foods. Findings from this study agreed with those reported by other researchers whereby overall acceptability scores were very high for porridges from Weanimix and OFSSoy [14]. The acceptability of OFSCri and OFSPal could be improved by repeated exposure and building a positive narrative towards the reemerging ancient practice of eating insects. Most of the sensory attributes were above the minimum threshold (hedonic scale = 3; neither like nor dislike) [21] of likeness for OFSCri and OFSPal, showing promise for their acceptance as CFs by Ghanaian mothers. However, OFSPal was found to be more acceptable to mothers than OFSCri. This is because mothers in the Volta region are more familiar with RFA [17] than crickets and its consumption has also been documented as an ancient practice among households in the region [16].

The findings of this research show that the OFSP-based edible insect CFs appear to be a promising alternative as an ASF for complementary feeding; however, we recognize some important limitations to this study. First, we did not measure the amount consumed either by the mother or child. Secondly, we did not report mother’s willingness to pay for or use these novel CFs for feeding their children. Although the Fe content of cricket and RFA is heme associated, it is not clear if its bioavailability will be sufficient to prevent Fe deficiency. We have attempted to describe the short-term nutritional and microbial profile of the OFSP-based edible insect CFs, but we did not assess the long-term stability and feasibility of home production of these novel CFs.

## 5. Conclusions

Using edible insects such as cricket or RFA, we have developed OFSP-based CFs that have the appropriate macronutrient and modest micronutrient contents for IYCF. These OFSP-based edible insect CFs are acceptable to mothers in urban and peri-urban settlements in the Volta region, Ghana. The OFSP-based edible insect CFs have several advantages over the mainstream CF (Weanimix) including ease of preparation, improved nutritional composition and lower risk of aflatoxin contamination. Because the ingredients are locally available and the production of these CFs is simple, they are likely to be a sustainable alternative ASF for IYCF. The OFSP-based edible insect CFs are safe for IYCF. However, future studies are needed in evaluating OFSCri and OFSPal’s infant acceptability and efficacy to have positive effects on linear growth and micronutrient status during the complementary feeding period. 

## Figures and Tables

**Figure 1 foods-09-01225-f001:**
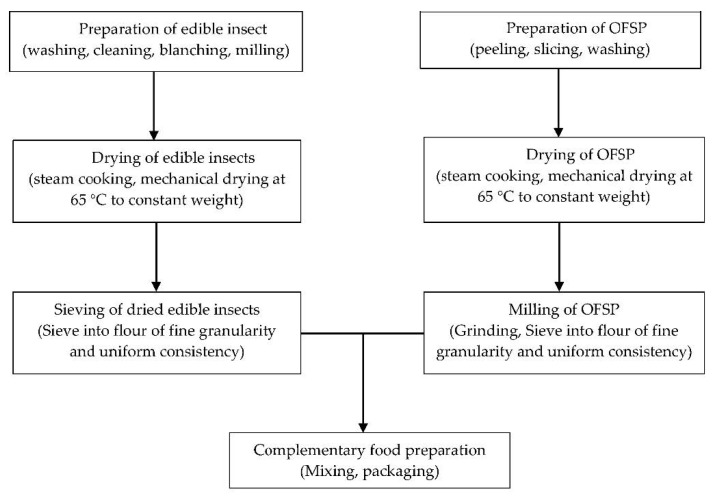
Preparation of orange–fleshed sweet potato (OFSP) complementary foods enriched with edible insects.

**Figure 2 foods-09-01225-f002:**
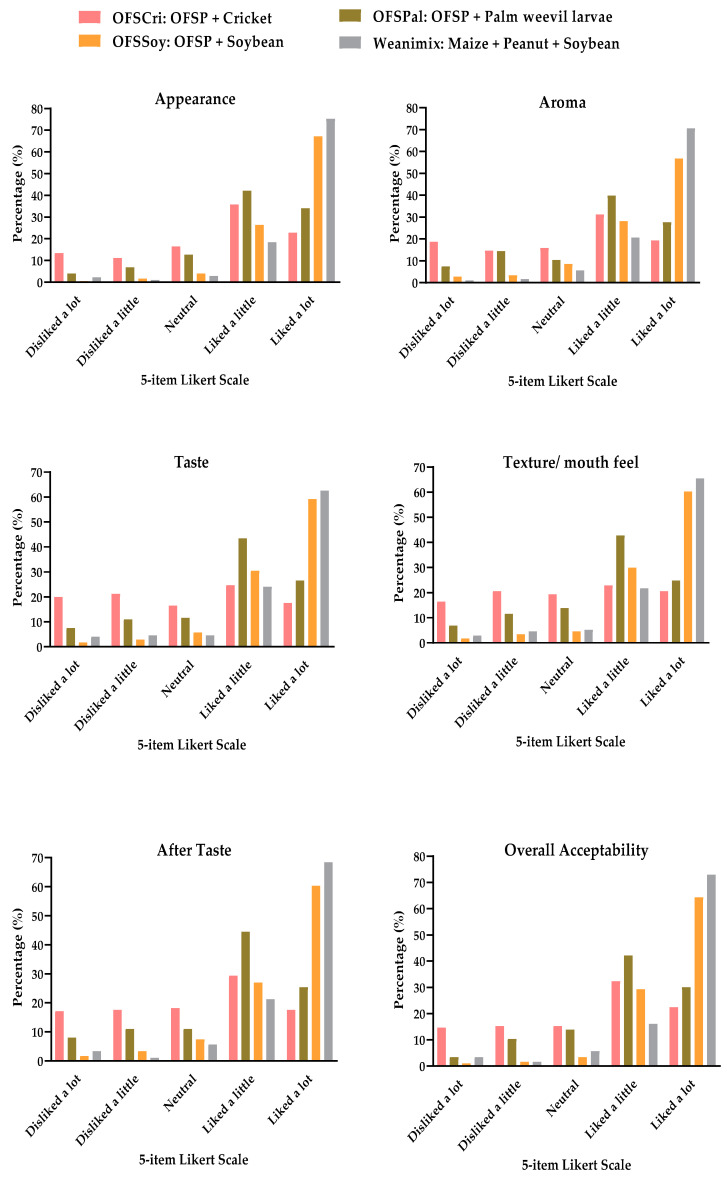
Maternal responses on sensory attributes of orange-fleshed sweet potato (OFSP) based complementary foods and Weanimix.

**Figure 3 foods-09-01225-f003:**
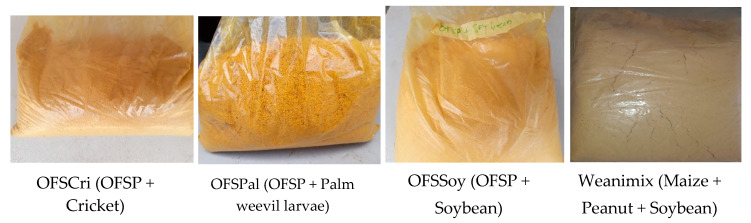
Color of orange-fleshed sweet potato (OFSP) based complementary foods and Weanimix.

**Table 1 foods-09-01225-t001:** Different proportions (%) of orange-fleshed sweet potato (OFSP), crickets, palm weevil larvae, soybeans, maize, and peanut used in the complementary foods.

Ingredient	OFSCri	OFSPal	OFSSoy	Weanimix
OFSP	70%	70%	70%	–
Cricket	30%	–	–	–
Palm weevil larvae	–	30%	–	–
Soybeans	–	–	30%	15%
Maize	–	–	–	75%
Peanut	–	–	–	10%

OFSP: Orange-fleshed sweet potato, OFSCri: OFSP + cricket; OFSPal: OFSP + palm weevil larvae; OFSSoy: OFSP + soybean, Weanimix: Maize-peanut-soybean blend.

**Table 2 foods-09-01225-t002:** Nutrient composition of orange-fleshed sweet potato based complementary foods and Weanimix on dry weight basis per 100 g.

Nutrient	OFSCri	OFSPal	OFSSoy	Weanimix	Standards
Moisture content ^1^, %	4.04 ± 0.59 ^a^	3.13 ± 0.21 ^b^	3.38 ± 0.32 ^c^	1.10 ± 0.15 ^d^	<5 ^α^
Energy, kcal	379.15 ± 10.83 ^a^	409.20 ± 9.06 ^b^	459.82 ± 10.99 ^c^	422.85 ± 3.29 ^d^	400 ^α^
Carbohydrate, g	64.00 ± 2.05 ^a^	73.16 ± 1.72 ^b^	57.09 ± 1.53 ^c^	60.05 ± 0.52 ^d^	60–75 ^α^*
Crude Protein, g	20.33 ± 0.58 ^a^	9.22 ± 0.20 ^b^	11.95 ± 0.29 ^c^	16.08 ± 0.13 ^d^	15 ^α^
Fat, g	1.24 ± 0.04 ^a^	6.93 ± 0.15 ^b^	17.39 ± 0.42 ^c^	9.91 ± 0.08 ^d^	10–25 ^α^
Crude Fiber, g	3.63 ± 0.27 ^a^	1.20 ± 0.13 ^b^	3.41 ± 0.45 ^a^	10.95 ± 0.16 ^c^	5 ^α^
Ash, g	6.76 ± 0.19 ^a^	6.36 ± 0.14 ^b^	6.78 ± 0.16 ^a^	1.91 ± 0.02 ^c^	-
Calcium, g	0.25 ± 0.01 ^a^	0.03 ± 0.00 ^b^	0.75 ± 0.02 ^c^	0.03 ± 0.00 ^b^	0.105 ^β^
Iron, mg	1.17 ± 0.03 ^a^	0.81 ± 0.02 ^b^	0.39 ± 0.01 ^c^	0.77 ± 0.01 ^b^	4.5 ^β^
Magnesium, g	2.25 ± 0.06 ^a^	2.26 ± 0.05 ^a^	3.26 ± 0.08 ^b^	1.81 ± 0.01 ^c^	*-*
Potassium, g	8.72 ± 0.25 ^a^	7.07 ± 0.16 ^b^	11.77 ± 0.28 c	7.05 ± 0.06 ^b^	*-*
Zinc, mg	0.13 ± 0.04 ^a^	0.14 ± 0.01 ^a^	0.05 ± 0.00 ^b^	0.19 ± 0.00 ^c^	1.6 ^β^

^1^ Moisture content not based on dry weight basis, OFSCri: Orange-fleshed sweet potato (OFSP) + Cricket, OFSPal: OFSP + Palm weevil larvae, OFSSoy: OFSP + Soybean, Weanimix: Maize + Peanut + Soybean. Within a row, means with different superscripts (letters) differ significantly (*p* < 0.05). ^α^ Source: Codex Alimentarius Commission [13], ^β^ Source: WHO recommended levels as stated by Dewey and Brown [22], * Estimated from data given for protein and fat in the codex standard.

**Table 3 foods-09-01225-t003:** Contribution (%) of energy, carbohydrate, protein, fat, Ca, Fe and Zn content of porridge from 100 g of OFSP-based CFs and Weanimx meeting RDA for children aged 6–12 months ^1^.

Nutrient	RDA ^δ^	OFSCri	OFSPal	OFSSoy	Weanimix
Energy, kcal	850 ^†^	44.6 ^a^	48.1 ^b^	54.1 ^c^	49.7 ^d^
Carbohydrate, g	95 *	67.4 ^a^	77.0 ^b^	60.1 ^c^	63.2 ^d^
Protein, g	11	184.8 ^a^	83.8 ^b^	108.6 ^c^	146.2 ^d^
Fat, g	30 ^#^	4.1 ^a^	23.1 ^b^	58.0 ^c^	33.0 ^d^
Calcium, mg	260 *	96.2 ^a^	11.5 ^b^	288.5 ^c^	11.5 ^b^
Iron, mg	11	10.6 ^a^	7.4 ^b^	3.5 ^c^	7.0 ^b^
Zinc, mg	3	4.3 ^a^	4.7 ^a^	1.7 ^b^	6.3 ^c^

^1^ RDA: Recommended dietary allowance, CFs: Complementary foods, OFSCri: Orange-fleshed sweet potato (OFSP) + Cricket, OFSPal: OFSP + Palm weevil larvae, OFSSoy: OFSP + Soybean, Weanimix: Maize + Peanut + Soybean. Within a row, means with different superscripts (letters) differ significantly (*p* < 0.05). ^δ^ Dietary Reference Intakes (2002/2005), * Adequate intake, ^#^ Fat based on acceptable macronutrient distribution range (AMDR) of 30% of energy [19], ^†^ Food and Nutrition Board [23].

**Table 4 foods-09-01225-t004:** Microbial quality of orange-fleshed sweet potato based complementary foods and Weanimix.

Sample	APC (cfu/g)	Mold & Yeast	*Bacillus cereus*	Enterobacteria	Fungi	*Salmonella*
OFSCri	38 × 10^1^	Mold = 0 Yeast = 0	0	0	0	None Detected
OFSPal	89 × 10^2^	Mold = 30 Yeast = 0	0	0	0	None Detected
OFSSoy	127 × 10^3^	Mold = 0 Yeast = 0	0	0	0	None Detected
Weanimix	1 × 10^1^	Mold = 0 Yeast = 0	0	0	0	None Detected
FDA standard ^†^	1 × 10^3^–1 × 10^4^	Not specified	1 × 10^2^–1 × 10^3^	1 × 10^1^–1 × 10^2^	Not specified	0

APC: Aerobic plate count, OFSCri: Orange-fleshed sweet potato (OFSP) + Cricket, OFSPal: OFSP + Palm weevil larvae, OFSSoy: OFSP + Soybean, Weanimix: Maize + Peanut + Soybean, ^†^ FDA: Ghana Food and Drugs Authority standard for dried products requiring heating to boiling before consumption.

**Table 5 foods-09-01225-t005:** Sensory evaluation of orange-fleshed sweet potato based complementary foods and Weanimix.

Attribute	OFSCri	OFSPal	OFSSoy	Weanimix	Score *
Appearance	3.44 ± 1.32 ^a^	3.95 ± 1.06 ^b^	4.58 ± 0.71 ^c^	4.63 ± 0.80 ^c^	4.15 ± 1.11
Aroma	3.18 ± 1.40 ^a^	3.66 ± 1.24 ^b^	4.33 ± 0.98 ^c^	4.58 ± 0.78 ^c^	3.96 ± 1.24
Taste	2.99 ± 1.41 ^a^	3.71 ± 1.19 ^b^	4.43 ± 0.86 ^c^	4.37 ± 1.04 ^c^	3.83 ± 1.28
Texture/mouth feel	3.11 ± 1.39 ^a^	3.67 ± 1.17 ^b^	4.44 ± 0.87 ^c^	4.43 ± 0.99 ^c^	3.88 ± 1.22
After taste	3.13 ± 1.36 ^a^	3.68 ± 1.20 ^b^	4.41 ± 0.90 ^c^	4.50 ± 0.92 ^c^	3.90 ± 1.24
Overall acceptance	3.32 ± 1.37 ^a^	3.85 ± 1.07 ^b^	4.54 ± 0.75 ^c^	4.53 ± 0.94 ^c^	4.03 ± 1.18

OFSCri: Orange–fleshed sweet potato (OFSP) + Cricket, OFSPal: OFSP + Palm weevil larvae, OFSSoy: OFSP + Soybean, Weanimix: Maize + Peanut + Soybean. Within a row, means with different superscripts (letters) differ significantly (*p* < 0.05). * Mean score for each sensory attribute across the various complementary foods.

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
