# Peer review of "Nutritional, Microbial, and Sensory Evaluation of Complementary Foods Made from Blends of Orange-Fleshed Sweet Potato and Edible Insects"

_foods, 2020, doi:10.3390/foods9091225_

Round 1

Reviewer 1 Report

The manuscript is a good contribution to the topic, it presents new data, and the theme is carefully elaborated.

However in my opinion the Results and Discussion sections were too long. It's good that everything is discussed, but sometimes it could be done in a more summary way so that it is easier for the reader to understand.

An important addition would also be to analyse the possibility of introducing the tested preparations into production and sale, taking into account the legal status of insects in Ghana, as to whether the proposed products would be easy to use in child nutrition (in Europe use insects as food is sometimes difficult).

Some minor flaws:

Line 73 Fabricius

Table 2 Check the table values

Since carbohydrates were calculated using the conversion method not in all columns of the table the sum of constituents is 100

Author Response

Response to Reviewer 1 Comments

Point 1: The manuscript is a good contribution to the topic, it presents new data, and the theme is carefully elaborated. However, in my opinion the Results and Discussion sections were too long. It's good that everything is discussed, but sometimes it could be done in a more summary way so that it is easier for the reader to understand.

Response 1: Results description have been reduced from 41/2 pages to 4 pages. Discussion has been reduced from 31/2 pages to 3 pages.

Point 2: An important addition would also be to analyse the possibility of introducing the tested preparations into production and sale, taking into account the legal status of insects in Ghana, as to whether the proposed products would be easy to use in child nutrition (in Europe use insects as food is sometimes difficult).

Response 2: The feasibility of introducing these foods into large scale production is beyond the scope of this study. However, this study provides a useful basis for decisions by small scale enterprises to scale up the production of these insect-based complementary foods. We have also reported on the willingness of the mothers to pay for these novel insect-based products in another paper hence cannot be included in this paper. In the willingness to pay manuscript, we also discussed the legal implications of producing and selling insect foods in Ghana, hence we cannot include it in this study. Indeed, these are important comments that have been addressed in a different manuscript. We have therefore acknowledged some of these comments in the limitations to the study on Lines 440-441.

Point 3: Some minor flaws: Line 73 Fabricius

Response 3: This has been changed on Line 80.

Point 4: Table 2 Check the table values. Since carbohydrates were calculated using the conversion method not in all columns of the table the sum of constituents is 100

Response 4: The carbohydrate content has been corrected in Table 2 and the sum of the constituents is 100. The corresponding carbohydrate percentage recommended dietary allowance met has also be corrected in Table 3.

Reviewer 2 Report

The study addresses an important topic and was very well conducted. All components of the manuscript were clearly and thoroughly written. I have just a few minor suggestions/clarifications.

For the maternal sensory evaluation, indicate if the mothers were blinded to which porridge they were tasting.

Lines 349-355: What amount of porridge is it presumed the children would consume to obtain these intake levels of energy, etc.? Are those amounts known to be the average amounts usually consumed by children in the age groups?  

Line 421: Do you mean "assess" instead of "access"?

Author Response

Response to Reviewer 2 Comments

Point 1: The study addresses an important topic and was very well conducted. All components of the manuscript were clearly and thoroughly written. I have just a few minor suggestions/clarifications. For the maternal sensory evaluation, indicate if the mothers were blinded to which porridge they were tasting.

Response 1: Mothers were blinded, and this has been indicated on Lines 219-220.

Point 2: Lines 349-355: What amount of porridge is it presumed the children would consume to obtain these intake levels of energy, etc.? Are those amounts known to be the average amounts usually consumed by children in the age groups?  

Response 2: The amount of porridge needed to meet the intake levels of energy have been specified and compared to usual amounts consumed by children. Refer to Lines 355-357.

Point 3: Line 421: Do you mean "assess" instead of "access"?

Response 3: The word “access” has been changed to “assess” on Line 421.

Reviewer 3 Report

The manuscript on "Nutritional, Microbial, and Sensory Evaluation of Complementary Foods Made from Blends of Orange-fleshed Sweet Potato and Edible Insects" is quite innovative towards improving the protein content of porridges for infant and young children in Ghana. Some important comments on the paper:

L43-45: A few examples of these plant-based complementary foods with low protein and micronutrient density.

L60-61: examples of these recommended indigenous plant and animal foods

It is not clear from Fig. 1 if OFSPal was de-fatted prior to milling? There may be concerns with rancidity during storage of the OFSPal complementary food.

Perhaps it could have been appropriate to also include a session for trained panelists in addition to the maternal sensory evaluation. This may help for future research on product formulation. Are there ways to enhance the color of OFSP insect-based complementary foods to improve their acceptability by mothers?

Author Response

Response to Reviewer 3 Comments

Point 1: L43-45: A few examples of these plant-based complementary foods with low protein and micronutrient density.

Response 1: Examples of the plant-based complementary foods have been indicated on Lines 46-47.

Point 2: L60-61: examples of these recommended indigenous plant and animal foods

Response 2: Examples of the recommended indigenous plant and animal foods have been indicated on Lines 66-67.

Point 3: It is not clear from Fig. 1 if OFSPal was de-fatted prior to milling? There may be concerns with rancidity during storage of the OFSPal complementary food.

Response 3: OFSPal was not defatted prior to milling. In figure 1, the fresh larvae were milling and dried thoroughly to reduce the fat and moisture content. The sieving also reduced the particle size of the dried larvae and hence the fat content per unit area. Although shelf life studies are needed to determine the effects of rancidity on OFSPal, our short-term storage conditions show that there were no concerns with rancidity during storage of OFSPal. Also, since OFSPal requires cooking before eating, small changes due to rancidity may have little to no impact on the quality of OFSPal. A summary of these explanations can be found on lines 95-98.

Point 4: Perhaps it could have been appropriate to also include a session for trained panelists in addition to the maternal sensory evaluation. This may help for future research on product formulation. Are there ways to enhance the color of OFSP insect-based complementary foods to improve their acceptability by mothers?

Response 4: Trained panels are used to determine treatment differences in the product being tasted. The objectives for training panelists are to inform the panel member on the test procedure, advance the individual’s ability to distinguish sensory characteristics, and improve the panel member’s memory resulting in consistent sensory judgments. The focus of this study was not to determine treatment differences in the products but to determine maternal acceptability. Hence consumer panels were used to test if the product will be accepted by the mothers. It is true that trained panelist may help with future research on product formulation, but we did not consider its use in this study due to the different foods that were being tested. We are optimistic that, the relatively low acceptability of the OFSP-based insect complementary foods is not due to the colour of the products but due to the aversion of eating insects as has been reported by previous researchers in Europe. We have included a sentence on way to improve the relatively low acceptability of the OFSP-based insect complementary foods. This can be found on lines 429-431.

Reviewer 4 Report

34. Explain complementary feeding period

40-41. Anemia worsens... Rewrite this sentence. It is not very clear what the authors are explaining

47-48 ASFs do not... Add the reference for this statement 

51-52 Edible insects.. Add the reference for this statement 

53 Add a comma after the as food, (Try to rewrite this sentences. It is too long and hard to understand. Break into 2 sentences). 

56 Using 1-2 sentences explain how the livestock impacts the environment

The introduction should be improved. Comment on why the crickets, palm weevil larvae and other consumed insects are more popular and how they began the consumption on insects. Why other insects are not consumed etc. Add a brief paragraph of this information so that it would be more interesting and informative for the readers. 

86. Avoid using previously described as there is not method described previously. Use "as described by".. or something similar to that 

In methods add the importance of each step. Why steam for 5 mins? Why dry for 24-48 hours etc. 

128. Add the reference of the dietary recommendation source 

135 Explain as to why these concentrations (Table 1) were chosen? Why not 50%-50% or 60%-40% 

151. Wrong degrees Celsius sign - correct it 

152. Made up to the mark is not scientific. Instead use topped up 

Avoid using the word "briefly" often 

200. The sensory evaluation has to be explained in more detail and heavily improved. How the session was briefed to them?, did the participants sign a consent form?,the lighting of the place where the test was conducted- since this would affect the appearance, how many participants?, were the data collected on paper?, Mothers not being needed to eat the whole amount- will this affect the results?

210. How can a score of 3 be selected as a threshold for acceptance? It can be argued that it could be a point of rejection. Are there any reasons to set the threshold to 3? Please explain. 

223. "met this requirement" doesn't sound scientific. Change the sentence

231. add "significantly" after differ

238. Add units for (18)

Table 3. Conduct statistical analysis to identify if the contributions are significantly different across different types of products per nutrient.  

302-307. Add the respective number from the scale in brackets after likes little, liked a lot, liked very much etc. 

318. It would be better to add the images of prepared porridge

as that was what the mothers ranked their ranking of appearance on. 

321. Add the source/ reference. 

324. Add the source/ reference. 

339. Previous should be corrected as previously.

355. Add the source/ reference. 

372. Add the source/ reference. 

426-436. Explain how you will overcome the challenge of low acceptability of OFSCri and OFSPal. 

There are a lot of instances where statements have been made without properly citing the previous literature. Add citations/ references. Check the whole manuscript. 

Author Response

Response to Reviewer 4 Comments

Point 1: L34. Explain complementary feeding period

Response 1: The complementary feeding period has been explained on Lines 34-36.

Point 2: L40-41. Anemia worsens... Rewrite this sentence. It is not very clear what the authors are explaining

Response 2: This sentence has been revised on Lines 42-43.

Point 4: L47-48 ASFs do not... Add the reference for this statement 

Response 4: Reference has been added on Line 51.

Point 5: L51-52 Edible insects. Add the reference for this statement 

Response 5: Reference has been added on Line 56.

Point 6: L53 Add a comma after the as food, (Try to rewrite this sentence. It is too long and hard to understand. Break into 2 sentences). 

Response 6: Sentence has been rewritten into two sentences on Lines 56-59.

Point 7: L56 Using 1-2 sentences explain how the livestock impacts the environment

Response 7: This explanation has been included on Lines 59-61.

Point 8: L34. The introduction should be improved. Comment on why the crickets, palm weevil larvae and other consumed insects are more popular and how they began the consumption on insects. Why other insects are not consumed etc. Add a brief paragraph of this information so that it would be more interesting and informative for the readers. 

Response 8: Introduction has been improved based on comments by this reviewer and from the other reviewers. Comments on why the crickets, palm weevil larvae and other edible insects are more popular is already in the introduction on Lines 56-64 and on Lines 77-84. Though we agree with the reviewer that providing more information about insects would be more interesting, we rather think that focusing on the nutritional value of the insects in improving the nutrient density of complementary and household foods is more appropriate and reflects the main objective of this study. 

Point 9: L86. Avoid using previously described as there is not method described previously. Use "as described by".. or something similar to that 

Response 9: This change has been corrected on Line 93.

Point 10: In methods add the importance of each step. Why steam for 5 mins? Why dry for 24-48 hours etc. 

Response 10: why the products were steamed and dried for 24-48 hours have been indicated on Lines 95-98. Th importance of the sieving is also stated on line 99 and 104.

Point 11: L128. Add the reference of the dietary recommendation source 

Response 11: Reference has been added on Line 138.

Point 12: L135 Explain as to why these concentrations (Table 1) were chosen? Why not 50%-50% or 60%-40% 

Response 12: Justification for the different proportions for each formulation can be found on Lines 138-139.

Point 13: L151. Wrong degrees Celsius sign - correct it 

Response 13: This has been corrected on Lines 163.

Point 14: L152. Made up to the mark is not scientific. Instead use topped up 

Response 14: This has been corrected on Line 164.

Point 15: Avoid using the word "briefly" often 

Response 15: The use of the word “briefly” has been avoided through the manuscript.

Point 16: L200. The sensory evaluation has to be explained in more detail and heavily improved. How the session was briefed to them?, did the participants sign a consent form?,the lighting of the place where the test was conducted- since this would affect the appearance, how many participants?, were the data collected on paper?, Mothers not being needed to eat the whole amount- will this affect the results?

Response 16: Further details on the sensory evaluation has been provided on Lines 217-220 and on lines 224-227. Mothers not being needed to eat the whole amount will not affect the results of the sensory evaluation in any way since this is a common practice in sensory science (Stone and Sidel, 2004).

Stone H., Sidel JL. 2004. Sensory Evaluation Practices. 3rd ed. San Diego, CA. Elsevier Academic Press.

Point 17: L210. How can a score of 3 be selected as a threshold for acceptance? It can be argued that it could be a point of rejection. Are there any reasons to set the threshold to 3? Please explain. 

Response 17: A threshold of 3 was set based on a previous study and this has been referenced on line 228.

Point 18: L223. "met this requirement" doesn't sound scientific. Change the sentence

Response 18: Various scientific publications have used the word “met” to describe whether an individual or a diet is meeting the recommended dietary allowance or not. As such we do not think that the use of the phrase “met this requirement” is out of place. Besides the other reviewers did not comment on this. We therefore think that it is a valid scientific expression and has been used in the US 2015-2020 Dietary guidelines. Please refer to the reference below to see the use of the expression in a scientific publication.

Beasley, J. M., Deierlein, A. L., Morland, K. B., Granieri, E. C., & Spark, A. (2016). Is Meeting the Recommended Dietary Allowance (RDA) for Protein Related to Body Composition among Older Adults?: Results from the Cardiovascular Health of Seniors and Built Environment Study. The journal of nutrition, health & aging20(8), 790–796. https://doi.org/10.1007/s12603-015-0707-5

Point 19: L231. add "significantly" after differ

Response 19: This has been added on line 247.

Point 20: L238. Add units for (18)

Response 20: (18) was a typo and has been deleted.

Point 21: Table 3. Conduct statistical analysis to identify if the contributions are significantly different across different types of products per nutrient.  

Response 21: Statistical test have been conducted and represented in Table 3. 

Point 22: L302-307. Add the respective number from the scale in brackets after likes little, liked a lot, liked very much etc. 

Response 22: The respective frequencies have been indicated on Lines 309-311. Appropriate footnote has also been included on line 277.

Point 23: L318. It would be better to add the images of prepared porridge as that was what the mothers ranked their ranking of appearance on. 

Response 23: There was no differences in the colours of the flours and that of the prepared porridges. We decided to show the pictures of the flours because they were in transparent Ziploc bags compared to the porridges that were dished into very small cups.

Point 24: L321. Add the source/ reference. 

Response 24: Reference has been added on Line 326.

Point 25: L324. Add the source/ reference. 

Response 25: Reference has been added on Line 328.

Point 26: L339. Previous should be corrected as previously.

Response 26: This has been corrected on line 343.

Point 27: L355. Add the source/ reference. 

Response 27: This sentence has been rewritten and does not need reference in its current state.  Please refer to Lines 353-355.

Point 28: L372. Add the source/ reference. 

Response 28: Reference has been added on Line 376.

Point 29: L426-436. Explain how you will overcome the challenge of low acceptability of OFSCri and OFSPal. 

Response 29: How to overcome the low acceptability of OFSCri and OFSPal have been included on Lines 429-431.

Point 30: There are a lot of instances where statements have been made without properly citing the previous literature. Add citations/ references. Check the whole manuscript. 

Response 30: The whole manuscript has been doubled checked for statements made without properly citing previous literature.  These have been checked and the manuscript has been read thoroughly again to check for minor grammatical errors.